# Robust RGB-D SLAM Using Point and Line Features for Low Textured Scene

**DOI:** 10.3390/s20174984

**Published:** 2020-09-02

**Authors:** Yajing Zou, Amr Eldemiry, Yaxin Li, Wu Chen

**Affiliations:** 1Shenzhen Research Institute, The Hong Kong Polytechnic University, Shenzhen 518057, China; rick.zou@connect.polyu.hk (Y.Z.); yaxin.pu.li@connect.polyu.hk (Y.L.); 2Department of Land Surveying and Geo-Informatics, The Hong Kong Polytechnic University, Hong Kong 999077, China; amr.eldemiry@connect.polyu.hk

**Keywords:** RGB-D SLAM, line features, low textured scene, sliding-window

## Abstract

Three-dimensional (3D) reconstruction using RGB-D camera with simultaneous color image and depth information is attractive as it can significantly reduce the cost of equipment and time for data collection. Point feature is commonly used for aligning two RGB-D frames. Due to lacking reliable point features, RGB-D simultaneous localization and mapping (SLAM) is easy to fail in low textured scenes. To overcome the problem, this paper proposes a robust RGB-D SLAM system fusing both points and lines, because lines can provide robust geometry constraints when points are insufficient. To comprehensively fuse line constraints, we combine 2D and 3D line reprojection error with point reprojection error in a novel cost function. To solve the cost function and filter out wrong feature matches, we build a robust pose solver using the Gauss–Newton method and Chi-Square test. To correct the drift of camera poses, we maintain a sliding-window framework to update the keyframe poses and related features. We evaluate the proposed system on both public datasets and real-world experiments. It is demonstrated that it is comparable to or better than state-of-the-art methods in consideration with both accuracy and robustness.

## 1. Introduction

Visual simultaneous localization and mapping (SLAM) can estimate camera motion and reconstruct a 3D scene simultaneously, which makes it a core technique in applications such as augmented reality (AR), virtual reality (VR) and robot navigation. Compared with a monocular and stereo camera, an RGB-D camera can provide pixel-wise color and depth information and becomes a popular choice for real-time dense reconstruction [1].

RGB-D SLAM has merged and developed as a sequence. Since the proposal of Parallel Tracking and Mapping (PTAM) [2], many feature-based methods have been introduced for RGB-D reconstruction, i.e., RGB-D SLAM v2, RTAB-Map and ORB-SLAM2 [3,4,5]. These methods exploit geometry constraint by extracting and matching point features, i.e., FAST, ORB, Shi-Tomasi, SIFT and SURF [6,7,8,9]. However, in the low textured scene, they cannot provide reliable constraint because few points are extracted, and many of them are wrongly matched.

Despite low texture, most indoor scenes contain abundant high-level geometry primitives such as lines and planes, which can be fused to aid camera tracking. Lines have been widely applied in monocular and stereo SLAM systems [10,11,12,13,14,15,16] but attracted less attention in the RGB-D research field [17,18,19]. Moreover, the existing line-based methods exploit either 3D–2D line correspondences or 3D–3D line correspondences [17,18,19]. The two-dimensional (2D) line segment will be neglected if it has no corresponding depth measurements. On the other hand, [17,19] the depth measurements of the 2D line during pose optimization can be ignored. Partial line information is not utilized by these methods.

In this paper, we propose a robust and comprehensive method to estimate the pose of an RGB-D camera and reconstruct an indoor environment. The main contributions are as follows:We exploit both 3D and 2D line reprojection error with point reprojection error and build a novel cost function that utilizes more line information than the previous methods.We build a robust pose solver to solve the camera pose and apply the Chi-Square test to detect the wrong point and line correspondences.We maintain a sliding-window framework to correct the camera pose and the related point and line features.We evaluate the proposed system on a public dataset and real-world experiments. Compared with state-of-the-art RGB-D SLAM systems, the proposed system can yield same-level accuracy in a common scene, and higher accuracy and robustness in low textured scenes.

In the rest of the paper, the related works are summarized in Section 2 and an overview of the proposed system is given in Section 3. Section 4 and Section 5 describe the frontend and backend of the proposed system, respectively. Experiment results are shown in Section 6 and conclusions are made in Section 7.

## 2. Related Works

This paper focuses on combining line features to recover the pose of an RGB-D camera. In this section, the works about RGB-D SLAM and line-based SLAM are reviewed, respectively.

### 2.1. RGB-D SLAM

RGB-D SLAM can be classified into two groups: (i) direct methods that extract all the geometry or photometric information such as Kinect-Fusion, Elastic-Fusion and DVO-SLAM [20,21,22]; (ii) feature-based methods that extract and match features from color images such as RGBDSLAM v2, RTAB-Map and ORB-SLAM2 [3,4,5].

Kinect-Fusion is a masterpiece for direct methods that can estimate camera pose and reconstruct scenes on GPU in real time. The current depth frame is aligned to a global volumetric model, and its pose is estimated by a coarse-to-fine Iterative Closest Point (ICP) algorithm [21]. However, it is limited to small workspaces due to high computation cost, memory consumption and lacking loop closure. As to lower the computation cost by map representation and update, Whelan et al. [23] present Kintinuous using a shift volumetric map, and Nießner et al. [24] maintain a lightweight map by combining sparse volumetric grid and voxel hashing. Whelan et al. [22] propose Elastic-Fusion which can reduce tracking drift and ensure global consistency by a two-step strategy. Firstly, local model-to-model verification is applied to detect local loop closure. Secondly, randomized fern encoding is implemented to detect global loop closure. Kerl et al. [20] develop DVO-SLAM, which optimizes both photometric and depth errors from all the pixels and leads to low localization drift. BAD-SLAM [25] incorporates a fast bundle adjustment (BA) algorithm into direct methods in real time. The cost function for direct BA fuses the geometry and photometric errors from the surfels which are used for scene representation, and then it is optimized in a similar way to SFM [26].

Compared with direct methods, feature-based SLAM is relatively efficient as only partial information is utilized. Henry et al. [27] designed an early RGB-D mapping system, where FAST features and Calonder descriptors are utilized to build feature matches. It uses the bag-of-word method [28] to improve the speed of loop closure detection and applies sparse BA to improve the accuracy of optimization. Engelhard et al. [29] introduce a hand-held RGB-D SLAM system for indoor mapping. The basic pipeline includes SURF feature extraction and matching, ICP for pose estimation, and pose graph optimization for refining trajectory. Endres et al. [3] extend this work comprehensively with more types of features and map representation. It provides SURF, SIFT and ORB features and evaluates their accuracy, robustness and runtime on TUM datasets, which indicates ORB is the most suitable for real-time application [30]. Both point cloud and octree-based maps are provided for 3D reconstruction [31]. Mur-Artal and Tardos [5] propose ORB-SLAM2 that can handle monocular, stereo and RGB-D frames. It is the first work composed of three threads: camera tracking, local mapping and loop closing. The comprehensive backend is constructed by bundle adjustment and pose graph optimization and can lower trajectory drift significantly. Tang et al. [32] introduce a hybrid SLAM system handling 2D–2D, 3D–2D and 3D–3D point pairs, in which the initial camera pose is determined by ICP using 3D–3D point pairs and then refined using all the pairs. Dai et al. [33] develop Bundle-Fusion which applies a sparse-to-dense approach for global pose estimation. Coarse camera poses are obtained using sparse SIFT features and refined by combining dense photometric and geometric errors. Real-time mapping is achieved based on surface reintegration with GPU.

### 2.2. Line-Based SLAM

Handling low textured scenes is not a difficult task for direct methods as they do not rely on texture for feature extraction. However, most direct methods with an RGB-D camera except for [20] use a dense volume for frame-to-model alignment and scene representation. GPU is required for the volume update, which constrains the applications of these methods. Direct methods with a monocular camera [34,35] can also provide robust results in low textured scenes. A semidense map is used for scene representation in LSD-SLAM [34], which can lower the computation cost significantly and enable real-time performance on commercial CPU. Sparse points can be sampled from edges and weak intensity variations in DSO [35], which are available despite low texture. Geometric camera calibration is integrated with photometric camera calibration to improve the tracking performance.

On the other hand, low textured scenes are still hard for feature-based methods as there are insufficient point features. This paper aims to improve the robustness of feature-based methods by fusing line features. Recent line-based methods are then investigated in this subsection.

Lemaire and Lacroix [13] proposed a line-based monocular SLAM using an extended Kalman filter (EKF). The line is represented by a Plücker coordinate and is updated together with a camera pose stored in a vector state. Pumarola et al. [15] built PL-SLAM upon ORB-SLAM, which is a monocular SLAM system. The line is represented by endpoints on the line, and 2D endpoint-to-line error is fused with point reprojection error for pose estimation. Gomez-Ojeda et al. [10] extend it to a stereo version and apply line descriptors in the bag-of-words approach for loop closure detection. He et al. [11] developed a tightly coupled visual-inertial odometry fusing point and line features. The Plücker coordinate and orthonormal representation are applied to represent and update the 3D line in a sliding-window framework [36]. Li et al. [14] proposed Structure-SLAM, which decouples rotation and translation estimation. The rotation matrix is first computed from line features and surface normal using the Manhattan World assumption, and then the translation is calculated based on the reprojection models of point and line features. Monocular line-based methods cannot provide a real scale. Furthermore, the reconstruction quality with a monocular or stereo camera is lower than that using an RGB-D camera. Lu et al. [18] designed robust RGB-D odometry fusing both points and lines. It uses two endpoints to represent a 3D line. Three-dimensional (3D) points are sampled on the 3D line and used to build 3D point-to-line errors. Fu et al. [17] extended PL-SLAM to the RGB-D version. It also uses endpoints to parametrize the 3D line, and project the 3D line to the 2D line segment on an image. Both 2D point and line reprojection errors are exploited to recover camera trajectory. Zhou et al. [19] presented Canny-VO, which extract Canny edge features and calculate the camera pose based on 3D–2D edge alignment.

As the 3D line has four degrees of freedom (DoF), this paper adopts orthonormal representation to avoid overparametrization. Plücker coordinate with six DoFs is also applied as it can conveniently transfer with orthonormal representation. Inspired by the works using either the 3D line feature or 2D line feature, we fuse both 3D and 2D line reprojection error. The proposed system can output accurate camera pose and reliable 3D model in low textured scenes, owing to the reliable and abundant line features.

## 3. System Overview and Notation

In this section, we depict the system design briefly and introduce the notations of transformation matrix, point and line features.

### 3.1. System Overview

The proposed system is built upon the open-source visual-inertial system FLVIS [37]. It proposes a feedback/feedforward loop to fuse the data from IMU and stereo/RGB-D camera. To work in low textured scenes with only an RGB-D camera, we disabled the function for IMU processing and added specific support for line features.

As shown in Figure 1, the proposed system has two parts: frontend and backend. We maintain a feature map to store camera poses, points, and lines, which can be updated by both frontend and backend.
Frontend: The frontend has one thread for pose tracking. Firstly, point and line features are detected in the current frame and matched with the previous frame. Secondly, the 3D information of the matched features in the world coordinate is searched in the feature map. Thirdly, a robust pose solver is built based on point and line reprojection errors, and wrong matches are deleted by the Chi-Square test. Fourthly, the camera pose is outputted, and the 3D model is expanded. Finally, the keyframe decision is made based on the relative motion and matched features from the previous keyframe. The feature map will be updated if a new keyframe comes.Backend: The backend has two threads: local mapping and loop closing. In the local mapping thread, a sliding-window bundle adjustment is implemented to update the feature map when a new keyframe arrives. In the loop closing thread, firstly, the arrived keyframe is transferred to a word vector by the bag-of-word approach, and the loop candidate is detected by the word vector comparison. The loop candidate is then verified by a geometry test by a Random Sample Consensus (RANSAC) Perspective-n-Point (PnP). Finally, the loop closure is corrected by pose graph optimization.

### 3.2. Notations

#### 3.2.1. Camera Pose Representation

We assume that all the depth measurements have been calibrated and registered to the RGB camera frame, so only the RGB camera frame is considered for coordinate transformation. The world frame is defined as the initial frame of the RGB-D camera. Camera pose is defined as the transformation ***T*** between the world frame and the camera frame, and represented by the manifold on the Special Euclidean Group (SE(3)) [38]. For example, in Figure 2, the transformation matrix from the world frame *w* to the camera frame *c*_1_ is represented by Twc1:(1)Twc1= (Rwc13×3twc13×101×31)= (qwc1twc1) ∈SE(3)
where Rwc13×3∈SO(3) represents the rotation matrix from the world frame to the camera frame, qwc1 is the unit quaternion parameterization and twc13×1 represents the translation from the world frame to the camera frame.

#### 3.2.2. Point Representation

Two types of representations for point feature have been utilized in SLAM systems: (a) its 3D position in world frame; (b) its inverse depth from the first keyframe observing it. The second type can deal with large-depth scenes, but it involves keyframe pose and is more complicated to transform between different frames. The proposed system chooses the first type, which is more widely used.

We assume that the 2D pixel measurement of a 3D point $P_{i}$ is $p_{ic}={\left(u_{ic},v_{ic}\right)}^{T}$ and its depth measurement is $pd_{ic}$. When $P_{i}$ is observed by a new keyframe for the first time, we can recover its 3D position in the world frame $P_{iw}={\left(x_{iw}\;,\;y_{iw}\;,\;z_{iw}\;\right)}^{T}$ and add it to the feature map.

#### 3.2.3. Line Representation

We treat a straight line in the world frame as an infinite line and use both the Plücker coordinate and orthonormal representation for line parameterization. The Plücker coordinate is convenient for line transformation and projection, while orthonormal representation is compact with four DoFs.

As shown in Figure 3, the Plücker coordinate consists of two 3D vectors ***d*** and ***m*** can be initialized by two points on the 3D line.
(2)$${𝓛}_{jc}=\left(\begin{array}{c} E_{jc}\times S_{jc}\\ E_{jc}-S_{jc} \end{array}\right)=\left(\begin{array}{c} m_{jc}\\ d_{jc} \end{array}\right)$$
where ${𝓛}_{jc}$ is the Plücker coordinate of 3D line in camera frame, $S_{jc}$ and $E_{jc}$ are two points on the line, $m_{jc}$ is the normal of the plane constructed by the line and frame origin, and $d_{jc}$ is the line direction. The transformation and projection of the Plücker coordinate will be introduced in Section 4.

To avoid overparameterization problem caused by the Plücker coordinate with six parameters, orthonormal representation (U, W)∈SO(3)×SO(2) is applied. We simply give the convention between orthonormal representation and Plücker coordinate below, and the detail can be referred to [16,36].
(3)$$U=\;R\left(\varphi \right)=\left(\begin{array}{ccc} \frac{m}{||m||} & \frac{d}{||d||} & \frac{m\times d}{||m\times d||} \end{array}\right)$$
where U is a 3D rotation matrix and $\varphi =\;{\left(\varphi_{x}\;,\;\varphi_{y}\;,\;\varphi_{z}\;\right)}^{T}$ is the rotation vector.
(4)$$W=\left(\begin{array}{cc} cos\theta & \;-\;sin\theta \\ sin\theta & cos\theta \end{array}\right)=\frac{1}{\sqrt{{||m||}^{2}+{||d||}^{2}}}\;\left(\begin{array}{cc} ||m|| & \;-\;||d||\\ ||d|| & ||m|| \end{array}\right)$$
where ***W*** is the 2D rotation matrix and θ is the rotation angle.

We use $\psi \;={\left(\varphi^{T},\;\theta \right)}^{T}$ for the minimal representation during sliding window bundle adjustment. The Plücker coordinate of the 3D line can be transferred from optimized ψ by
(5)$$𝓛=\left(\begin{array}{cc} cos\theta u_{1}^{T} & sin\theta u_{2}^{T} \end{array}\right)$$
where ***u***_1_ and ***u***_2_ are the first and second column of U.

## 4. Frontend

In this section, we introduce the frontend pipeline in detail. The feature correspondences are built based on feature extraction and matching, and then sent to a robust pose solver, which can filter out outlier matches and output robust pose estimation.

### 4.1. Feature Extraction and Matching

We use Shi-Tomasi as the point feature extractor, which is improved based on the Harris corner [9]. As shown in Figure 4a, the image plane is divided into 16 regions and newly detected features are added to these regions based on the score of the Harris index [39]. The maximum number of features in every region is set as 30.

For the first frame, points are extracted from the color image and their depths are recovered from the depth image. These points are then added to the feature map as landmarks. For the following frames, these points are tracked by the Lucas–Kanade optical flow [40], and new points will be reextracted and selected from the 16 regions until the maximum number is reached.

A more uniform distribution of point features is achieved by dividing the image into smaller regions and controlling the number of features in these regions. We have tuned the value of region number and feature number in the region, and we argue that 16 regions with a maximum of 30 features are suitable for images with 640 × 480 resolution.

We use LSD as the line feature extractor as shown in Figure 4b, which can detect line segments with high accuracy and fast speed [41]. We extract the binary descriptor for the line segment using LBD, which is an efficient line descriptor with both appearance and geometry constraint [42].

The combination of LSD and LBD has been widely applied in line-based methods [10,14,15,17,18]. Though the computation cost increases owing to the extraction of LSD and LBD, robust performance allows for low textured scenes. To improve the speed, we control the number of the line features and set the maximum as 100. Line features are selected based on the length and distribution and those with small lengths or near the boundary of the image are less likely to be selected.

We combine the appearance information from LBD and the geometry information from LSD to match the line features from consecutive frames. A three-step method is detailed below:Cross-check. FLANN [43] is applied twice to match line descriptors. In the first matching, the descriptors from the previous frame are set as the query set, and those from the current frame are set as the train set. We can find a matched descriptor DesT_1_ in the train set for a descriptor DesQ_1_ in the query set. By contrast, in the second matching, the descriptors from the previous frame are set as the train set, and those from the current frame are set as the query set. Again, we can find DesT_2_ and DesQ_2_. DesT_1_ and DesQ_2_ are from the previous frame, while DesT_2_ and DesQ_1_ are from the current frame. If DesT_1_ and DesQ_2_ are the same descriptor, then DesT_2_ and DesQ_1_ should also have the same descriptor index. Otherwise, they will be removed as the wrong feature match.Ratio-test. We assume that DesT_1_ is the matched feature of DesQ_1_ after cross-check, which means DesT_1_ has the smallest distance from DesQ_1_ among the train set. We argue that the ratio between DesT_1_ and the second smallest distance should be smaller than 0.75. Otherwise, the feature match DesT_1_ and DesQ_1_ is rejected.Geometry test. We then associate DesT_1_ and DesQ_1_ with two line segments based on their indexes. LSD provides the orientation, length, and endpoints of the line segments. If the line segments have highly different orientations, length or endpoints, we will discard the line match and not use it for pose estimation.

For a new keyframe, if extracted line features are not matched to any line landmark in the feature map, we will recover their Plücker coordinates and store their information in the feature map.

### 4.2. Robust Pose Solver

In this subsection, we first introduce an infinite impulse response (IIR) filter to update the point landmark in the feature map. We argue that the depth measurement is fused into the updated point landmark by IIR filter, so we can neglect it and only use pixel measurements during pose optimization. We then derive the 2D point reprojection error, 2D and 3D line reprojection error. Finally, we combine them to build a novel cost function and detect the wrong feature matches based on the Chi-Square test.

#### 4.2.1. Infinite Impulse Response Filter

If the tracked point feature has reliable depth measurement, e.g., $pd_{ic}$ is larger than 0.2 m and smaller than 6.0 m, its measured position in camera frame is derived by
(6)$$P_{ic}^{measure}=\;pd_{ic}K^{\;-1}{\left(\begin{array}{ccc} u_{ic}, & v_{ic}, & 1 \end{array}\right)}^{T},\;K\;=\;\left(\begin{array}{ccc} f_{x} & 0 & c_{x}\\ 0 & f_{y} & c_{y}\\ 0 & 0 & 1 \end{array}\right)$$
where K is the intrinsic parameter matrix.

We can find its corresponding landmark in the feature map and project it from the world frame to the camera frame by
(7)Picproject= TwcPiw

The IIR filter is applied to update its position by
(8)$$P_{ic}=\lambda P_{ic}^{project}+\left(1-\lambda \right)\;P_{ic}^{measure}$$
where λ is the parameter of IIR filter. The advantage of the IIR filter is that it utilizes all the measurements of the point landmark throughout the lifespan. The position error of the landmark will converge faster, and the negative effect of depth outlier will be lowered. From our experience, the IIR filter works better if we set λ between 0.6 and 0.9. The value of λ indicates the confidence on the historical information in the feature map, while the value of 1−λ means the confidence of the quality of the feature extractor and depth measurements.

#### 4.2.2. Two-Dimensional (2D) Point Reprojection Error

If point $P_{iw}$ is tracked to camera frame, and the pixel measurements on the image plane is $p_{ic}$, we can derive the 2D point reprojection error
(9)ric2p=pic−f(KTwcPiw), f(abc)=(a/cb/c)
where f is a normalization function.

#### 4.2.3. Three-Dimensional (3D) Line Reprojection Error

We can transform Plücker coordinate ${𝓛}_{jw}$ from world frame to camera frame by
(10)𝓛jcproject=(mjcprojectdjcproject)(Rwctwc∧Rwc0Rwc) 𝓛jw

We then sample and project points on the 2D line segments to 3D space. If more than 70% of these points have reliable depth measurements, we argue the depth measurements along the 2D line segment are reliable. We can robustly fit the projected points to Plücker coordinate ${𝓛}_{jc}$.

Because both ${𝓛}_{jc}^{project}$ and ${𝓛}_{jc}$ have six parameters which are overparameterized, we do not build a 3D line reprojection error based on the Plücker coordinate difference. Instead, we transfer them to orthonormal representation $\psi_{jc}^{project}$ and $\psi_{jc}$ for compact comparison by Equations (4) and (5). The 3D line reprojection error is
(11)$$r_{jc}^{3l}=\;\psi_{jc}-\;\psi_{jc}^{project}$$

#### 4.2.4. Two-Dimensional (2D) Line Reprojection Error

If the depth measurements along the matched 2D line segment are not reliable, we can project the Plücker coordinate ${𝓛}_{jc}^{project}$ from the camera frame to the image plane by
(12)$$l_{jc}^{project}=\left(\begin{array}{c} l_{1}\\ l_{2}\\ l_{3} \end{array}\right)=𝓴m_{jc}^{project}=\left(\begin{array}{ccc} f_{y} & 0 & 0\\ 0 & f_{x} & 0\\ -f_{y}c_{x} & -f_{x}c_{y} & f_{x}f_{y} \end{array}\right)m_{jc}^{project}$$
where $m_{jc}^{project}$ is the plane normal of ${𝓛}_{jc}^{project}$, $l_{jc}^{project}$ is the 2D projected line and 𝓴 is the line projection matrix.

As shown in Figure 3, the 2D line reprojection error is defined as the distance from endpoints to the projected line $l_{jc}^{project}$
(13)$$r_{jc}^{2l}={\left(\begin{array}{cc} r_{jc}^{2s}, & r_{jc}^{2e} \end{array}\right)}^{T}=\;{\left(\begin{array}{cc} \frac{u_{jc}^{s}l_{1}+v_{jc}^{s}l_{2}+l_{3}}{\sqrt{l_{1}^{2}+l_{2}^{2}}}, & \frac{u_{jc}^{e}l_{1}+v_{jc}^{e}l_{2}+l_{3}}{\sqrt{l_{1}^{2}+l_{2}^{2}}} \end{array}\right)}^{T}$$
where $s_{jc}=\;{\left(\begin{array}{cc} u_{jc}^{s}, & v_{jc}^{s} \end{array}\right)}^{T}$ and $e_{jc}=\;{\left(\begin{array}{cc} u_{jc}^{e}, & v_{jc}^{e} \end{array}\right)}^{T}$ are the endpoints of the detected line segment on the image plane.

#### 4.2.5. Novel Cost Function

We combine Equations (9), (11) and (13) to build a novel cost function below
(14)$$\underset{i}{\sum }\rho \left(||r_{ic}^{2p}{||}_{\Sigma_{ic}^{2p}}^{2}\right)+\underset{j}{\sum }\rho \left(||r_{jc}^{3l}{||}_{\Sigma_{jc}^{3l}}^{2}\right)+\underset{j}{\sum }\rho \left(||r_{jc}^{2l}{||}_{\Sigma_{jc}^{2l}}^{2}\right)$$
where ρ is the Huber function and Σ is the covariance matrix associated with the reprojection error. The iterative Gauss–Newton method implemented in g2o is applied to minimize the cost function and solve Twc [44]. The details are as below:The initial camera pose is solved by the RANSAC PnP method before minimizing the function. Wrong matches among 2D features are filtered out by RANSAC. If the relative motion between the previous frame and the initial camera pose exceeds a threshold, the initial guess from RANSAC PnP will be rejected and is calculated again using a constant-velocity motion model. We directly use the implementation of RANSAC PnP from OpenCV [45]. After 100 iterations, the point feature will be removed if its reprojection error is larger than three pixels. We assume that the camera motion during a short period follows a constant-velocity assumption, so we can predict the camera pose of the current frame using the velocity and camera pose of the previous frame.For all the line matches, the 2D line reprojection error is calculated based on the initial camera pose and 3D line landmarks in the feature map using Equation (13). The line matches associated with large initial line reprojection errors are filtered out.The remaining feature matches are then sent to Equation (14) for optimization. After every four iterations, we filter out the matches that fail in the Chi-Square test and continue to optimize using the remaining matches.
(15)$$||r_{ic}^{2p}{||}_{\Sigma_{ic}^{2p}}^{2}<\;\chi {\;}_{\alpha ,n_{2p}}\;,||r_{jc}^{3l}{||}_{\Sigma_{jc}^{3l}}^{2}<\;\chi {\;}_{\alpha ,n_{2l}},||r_{jc}^{2l}{||}_{\Sigma_{jc}^{2l}}^{2}<\;\chi {\;}_{\alpha ,n_{3l}}$$
where α the threshold of Chi-Square test, and $n_{2p}=2$ ,$\;n_{3l}=4$ and $n_{2l}=2$ are the degrees of freedom associated with the reprojection error.

The analytical Jacobian matrices of line reprojection errors with respect to camera pose can be referred to [36,46]. The proposed system uses automatic differentiation in g2o to compute the Jacobian matrices [44].

## 5. Backend

This section introduces the backend of the proposed system, with two parallel threads, local mapping and loop closing.

### 5.1. Local Mapping Thread

As shown in Figure 5, the frontend will publish a keyframe message if a new keyframe is determined by the relative motion from the previous keyframe. For example, if the relative translation exceeds 0.1 m, or the relative rotation angle exceeds 0.2 rad, we argue that the camera has moved enough, and the feature map needs to be updated by a new keyframe.

The keyframe message contains the camera pose, and point and line associations attached to the current frame. If the keyframe message arrives in the local mapping thread, it will delete the oldest keyframe and add the new keyframe to the sliding-window framework to fix the keyframe number. As well, the features that are observed only by the oldest keyframe will be deleted accordingly.

The novel cost function in the sliding-window is presented as
(16)$$\underset{k}{\sum }\underset{i}{\sum }\rho \left(||r_{ik}^{2p}{||}_{\Sigma_{ik}^{2p}}^{2}\right)+\underset{k}{\sum }\underset{j}{\sum }\rho \left(||r_{jk}^{3l}{||}_{\Sigma_{jk}^{3l}}^{2}\right)+\underset{k}{\sum }\underset{j}{\sum }\rho \left(||r_{jk}^{2l}{||}_{\Sigma_{jk}^{2l}}^{2}\right)$$
where *k*, *i* and *j* are the indexes of keyframe poses, points and lines, respectively. The oldest keyframe pose is set fixed, and all the other keyframe poses and features (i.e., Twk, $P_{iw}$ and $\psi_{jw}$) will be refined by the iterative Gauss–Newton method in g2o. The Chi-Square test is applied again to filter out wrong matches from the frontend. Finally, the local mapping thread will publish a correction message to the frontend. The pose of the current frame and related features will be updated accordingly.

### 5.2. Loop Closing Thread

The keyframe message is also sent to the loop closing thread, which has three parts, loop closure detection, loop closure verification and pose graph optimization.

#### 5.2.1. Loop Closure Detection

DBoW2 is applied to detect loop candidates, which is a bag-of-word approach [47]. DBoW2 has been widely applied in the SLAM system for loop detection and shows the advantages of speed and accuracy [5,48].

We redetect ORB features and extract ORB descriptors from the new keyframe and transfer the descriptors to a word vector. The loop candidate is determined by the similarity score between the word vector of the new keyframe and the previous keyframe. Four conditions are set to find out the loop candidate keyframe:The difference between the indexes of the new keyframe and candidate keyframe exceeds 100. Therefore, a keyframe close to the new keyframe will not be selected.The candidate keyframe has the highest score.The highest score should exceed 0.15. Otherwise, we argue that the similarities between the two keyframes are insufficient.The scores of continuous three keyframes before the candidate keyframe exceed 0.12. Thus, an isolated keyframe with the highest score will be rejected, which may be caused by perceptual aliasing.

#### 5.2.2. Loop Closure Verification

Wrong loop closure may occur due to perceptual aliasing, especially when the surveying environment contains a similar texture. Therefore, the loop candidate should be verified by geometry constraint. To build correspondences between the new keyframe and the candidate frame, FLANN is applied to associate their ORB descriptors. The wrong matches are ruled out by cross-check and ratio-test first. Then the relative motion between the new keyframe and the loop candidate is calculated by RANSAC PnP. The loop candidate will be rejected if insufficient inlier matches are found or relative motion exceeds a threshold.

#### 5.2.3. Pose Graph Optimization

Pose graph optimization will be performed to correct the loop closure if the loop candidate is verified by geometry constraint. For the pose graph optimization, the vertexes are the keyframe poses, and the edges are the transformation between adjacent keyframes, and that between loop keyframes as below
(17)rm,n=log(Twkm−1∗Tknkm∗Twkn)
where *km* and *kn* are the indexes of the two keyframes associated with the edge, and log is to convert the transformation matrix to its Lie Algebra.

The cost function of pose graph optimization is built as
(18)$$\underset{a}{\sum }\rho \left(||r^{a,a+1}{||}_{\Sigma^{a,a+1}}^{2}\right)+\underset{l}{\sum }\rho \left(||r^{l1,l2}{||}_{\Sigma^{l1,l2}}^{2}\right)$$
where $r^{a,a+1}$ is the transformation error between adjacent keyframes and $r^{l1,l2}$ is the error between loop keyframes. The loop closure will be corrected by the iterative Gauss–Newton method in g2o, and keyframe poses will be refined accordingly.

## 6. Experiments and Discussion

The system performance is evaluated by TUM RGB-D datasets and real-world experiments and compared with state-of-the-art systems [30]. The experiments of the proposed system are carried out on a standard laptop (CPU: Core i5-5200U; RAM 8G).

### 6.1. Evaluation Matrix and Tool

The absolute trajectory error (ATE) is used to reflect the drift between ground truth trajectory and estimated trajectory. The root mean square error (RMSE) of ATE is applied to evaluate the system accuracy. The definitions of ATE and RMSE are shown below
(19)ei=trans(Tgti)− trans(TestiTgtest)
(20)$$\mathrm{RMSE}=\sqrt{\frac{\sum_{i=1}^{n}e_{i}^{T}e_{i}}{n}}$$
where Tgti is the ground truth for camera pose, Testi is the estimated pose from the SLAM system, and Tgtest is the transformation between two trajectories by Umeyama alignment [49].

The open-source package evo (https://michaelgrupp.github.io/evo/) is applied as the evaluation tool to compare the proposed system with state-of-the-art systems.

### 6.2. TUM RGB-D Datasets

TUM RGB-D datasets consist of sequences recorded with a Microsoft Kinect RGB-D camera in a variety of scenes. The frequency of the datasets is 30 frames per second (FPS), and the resolution is 640 × 480. The ground truth trajectory is recorded with a high-accuracy motion capture system with 100 Hz. Ten sequences are selected for trajectory evaluation. fr1_desk, fr1_floor, fr2_desk and fr3_long_office are common indoor scenes with texture and structure, fr3_nstr_tex_far and fr3_nstr_tex_near lack structure, fr3_str_ntex_far and fr3_str_ntex_near lack texture, and fr3_str_tex_far and fr3_str_tex_near contain highly discriminative texture.

We compare the performance of the proposed system with state-of-the-art systems, i.e., ORB-SLAM2, DVO-SLAM, LSD-SLAM, DSO, PL-SLAM and Canny-VO [5,15,19,20,34,35]. We evaluate PL-SLAM [15] using the implementation from https://github.com/HarborC/PL-SLAM, as its original code is not opens-sourced. The scales of trajectories from LSD-SLAM, DSO and PL-SLAM [15,34,35] are corrected by aligning to the ground truth trajectories. Table 1 shows the comparison results of ATE RMSE, where “-” represents tracking failure, and “X” means that the result is not provided in the related paper. The smallest values are bolded and indicate the best accuracy.

In Table 1, most of the systems can yield high accuracy (RMSE/Length < 1%) in most sequences. ORB-SLAM2 yields the best accuracy in 1 sequence out of 10. The proposed system, DVO-SLAM and PL-SLAM yield the best accuracy in two sequences, and Canny-VO achieves the highest accuracy in three sequences.

ORB-SLAM2 fails in fr3_str_ntex_far and fr3_str_ntex_near with low texture as it is highly dependent on the point feature. The performance of LSD-SLAM will degrade if the camera is too close to walls or floors, i.e., in fr3_str_ntex_near and fr3_str_tex_near. DSO delivers much worse results than its original paper [35], because the mono datasets in [35] prepare photometric calibration files while TUM RGB-D datasets do not. PL-SLAM is built based on the monocular version of ORB-SLAM2. It can outperform ORB-SLAM2 in some sequences owing to the fusion of line features, but its results are not so robust that four sequences are not successfully tracked. Canny-VO fails in fr3_str_ntex_near due to ambiguous structure.

The proposed system and DVO-SLAM success in all the sequences, excepting for fr1_floor which are not published [20]. DVO-SLAM is a dense SLAM system exploiting the photometric and depth information from all the pixels instead of point features, so it is still robust in low textured scenes. On the other hand, the robustness of the proposed system is from the fusion of both 3D and 2D line features. Therefore, we conclude that compared with state-of-the-art works, the proposed system yields a similar level of accuracy and high robustness in a variety of scenes.

### 6.3. Room Experiment

We record an experiment sequence with a Kinect v2 in a laboratory room with an installed motion capture system as shown in Figure 6. The ground truth of the trajectory is recorded by Qualisys [50]. The sequence covers some challenging scenes, i.e., low texture, illumination variation and glass window as shown in Figure 7. The main difficulty of the sequence is that the Kinect v2 will cross the low textured wall.

To further verify the improvement of localization accuracy by the fusion of comprehensive line features, we present five evaluations of the experiment sequence in Table 2: (a) ORB-SLAM2; (b) point; (c) point + 3D line; (d) point + 2D line and (e) the proposed system. In Evaluations (b)–(d), we disable some functions in the proposed system, so it runs with only point feature, point + 3D line feature, and point + 2D line feature, respectively. The RMSE of ATE is used to evaluate the localization accuracy again. The smallest values are bolded and indicate the best accuracy.

The results in Table 2 indicate the benefits of the proposed system. ORB-SLAM2 loses tracking when crossing the low textured wall, while other evaluations succeed. Considering the decrease of RMSE, the proposed system yields the best accuracy and can improve the accuracy of Evaluations (b)–(d) by 7.7 cm, 2.1 cm and 2.5 cm, respectively.

The trajectories of Evaluations (b)–(e) are shown in Figure 8. The red circle in Figure 8a indicates the partial trajectory crossing the low textured wall. The green circle in Figure 8d indicates the closed loop of the trajectory. The blue circle in Figure 8d indicates the end of the trajectory.

Due to lacking reliable and sufficient point features, Evaluation (b) has the highest trajectory error when crossing the wall, which is still significant after loop closing. For Evaluations (c)–(e), with the help of robust line features, the trajectory errors crossing the wall are much smaller. The values of the color bar in Figure 8 also indicates the accuracy improvement of the proposed system.

Figure 9 shows the reconstruction models from ORB-SLAM2 and the proposed system. The red circle in Figure 9a indicates the low textured wall where ORB-SLAM2 loses tracking. Therefore, only the partial trajectory is outputted, and the partial model is reconstructed by ORB-SLAM2. On the other hand, the proposed system can generate a complete 3D model owing to robust line fusion.

### 6.4. Corridor Experiment

We record an experiment sequence with an iPad and a structure sensor in a corridor. The ground truth of the corridor model is provided by NavVis M6, a laser-based indoor mapping system. Figure 10 shows the experiment device and the ground truth model. Referring to Figure 4, the corridor scene covers rich line features.

To further verify the improvement of mapping accuracy by the proposed system, we present five evaluations in Table 3: (a) ORB-SLAM2; (b) point; (c) point + 3D line; (d) point + 2D line and (e) the proposed system. We incrementally build a 3D corridor model based on the outputted camera pose and registered RGB-D frames and then compare the final reconstruction model with the ground truth. The reconstruction model and ground truth are aligned by Cloud-Compare and the RMSE of the point-to-point distance are used to evaluate the reconstruction quality [51].

As shown in Table 3, the smallest value is indicated in bold. The proposed system yields the highest mapping quality among all the evaluations.

For more intuitive comparison, the reconstruction models from five evaluations are shown in Figure 11. The color of the model indicates the value of the point-to-point distance. The red circles indicate the biggest point-to-point distances after the second turning of the corridor. The five circles in Figure 11a–e are of the same size, and we select the areas with the biggest distances to indicate the mapping quality of the five evaluations.

Obviously, Evaluation (e) has the smallest area, Evaluation (b) has the biggest area, while Evaluations (c)–(d) both improve and have smaller areas than Evaluation (b). Though the area of Evaluation (a) from ORB-SLAM2 is the second smallest, its point-to-point distance at the end of the corridor is higher than the rest evaluations, which increases the value of RMSE in Table 3. We can argue that fusing 3D or 2D line features can improve the mapping quality, and the combination of both 3D and 2D line features in the proposed system can yield the best mapping quality in the five evaluations.

We then investigate the computation speed of the proposed system. The time consumption by loop closure is not listed as it is highly dependent on the keyframe number. Similarly, the time cost of incremental mapping is also increasing with the frame number, so its processing time is not listed either. Table 4 shows the processing time of each part in the tracking thread and local mapping thread and compares the total time cost with ORB-SLAM2.

Due to the time consumption by line extraction, the tracking frequency of the proposed system is only half of ORB-SLAM2. We sacrifice computation speed for more robustness in low textured scenes. The sliding-window BA of the proposed system is much faster than the local BA of ORB-SLAM2 for two reasons:The number of keyframes of the proposed system is lower than ORB-SLAM2. While the proposed system maintains a sliding-window with eight keyframes, ORB-SLAM2 builds a covisibility map for every keyframe, where the connected keyframes can be more than 20.While ORB-SLAM2 needs to build a new optimizer using g2o for every keyframe, the proposed system maintains the same optimizer for every keyframe. Therefore, the optimizer of the proposed system can converge faster, and it also saves the time to build the new optimizer.

## 7. Conclusions

To overcome the significant drift of the SLAM system in low textured scenes, the paper presents a robust RGB-D SLAM system using point and line features. In conclusion,
To comprehensively fuse 3D and 2D line features, we build a novel cost function combining point reprojection error and 3D and 2D line reprojection error.We build a robust pose solver to solve the novel function and recover camera pose from the point and line matches. We maintain a sliding-window framework to update the camera poses and the corresponding landmarks.Experiment results of the TUM dataset show that the proposed system can achieve the same-level accuracy in common scenes compared with the state-of-the-art system and can improve the robustness in low textured scenes. The room experiment shows the improvement of localization accuracy of the proposed system compared with point-based SLAM, and also indicates the benefit of fusing both 3D and 2D lines. The corridor experiment further reflects the improved mapping quality of the proposed system owing to 3D and 2D line fusion.

In the future, we will extend this work from two aspects: (a) Exploiting the Manhattan World constraint from the 3D line features; (b) Combing plane features for more robustness in low textured scenes.

## Figures and Tables

**Figure 1 sensors-20-04984-f001:**
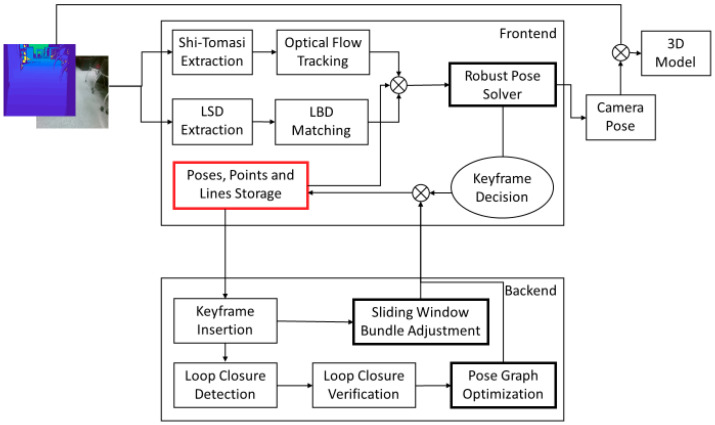
System overview. The input of the proposed system is the calibrated RGB and depth frames, and the output is the camera pose and 3D model. The proposed system has two parts: frontend and backend. We extract and match point and line features, and estimate the camera pose in the frontend. We apply sliding-window bundle adjustment to update camera poses, points and lines and apply loop closure correction to refine camera poses in the backend.

**Figure 2 sensors-20-04984-f002:**
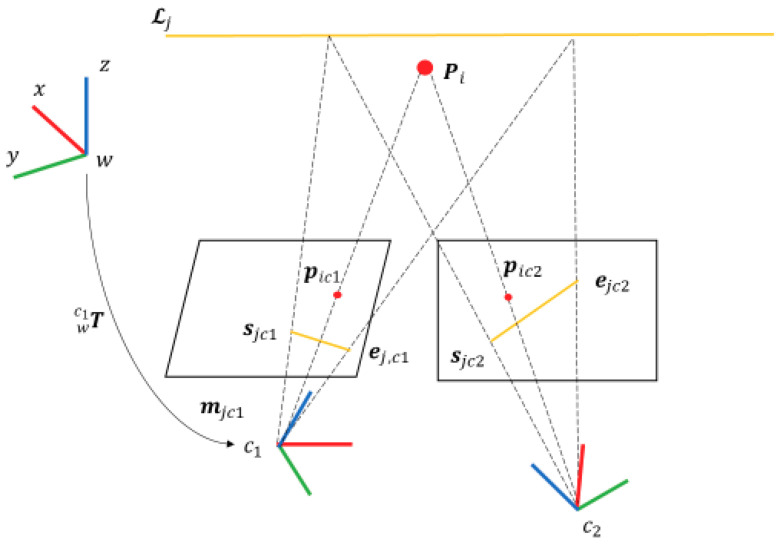
An illustration of the RGB-D camera and point and line measurements.

**Figure 3 sensors-20-04984-f003:**
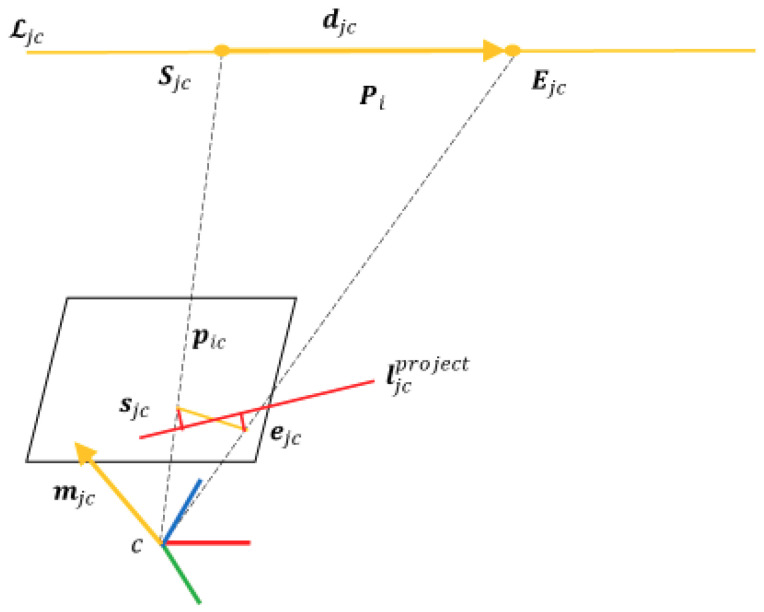
Plücker coordinate of a straight line.

**Figure 4 sensors-20-04984-f004:**
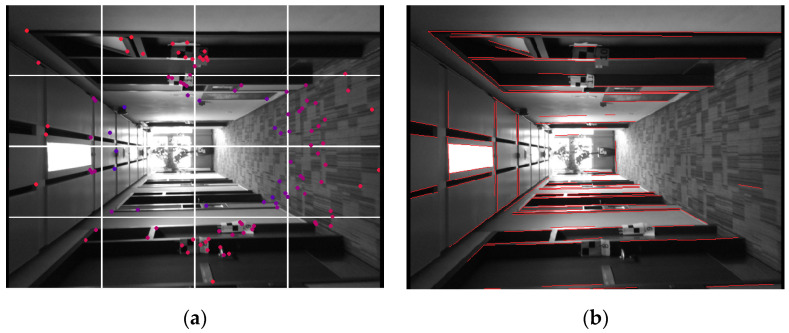
Point and line feature extractor. (**a**) Improved Shi-Tomasi extractor; (**b**) LSD.

**Figure 5 sensors-20-04984-f005:**
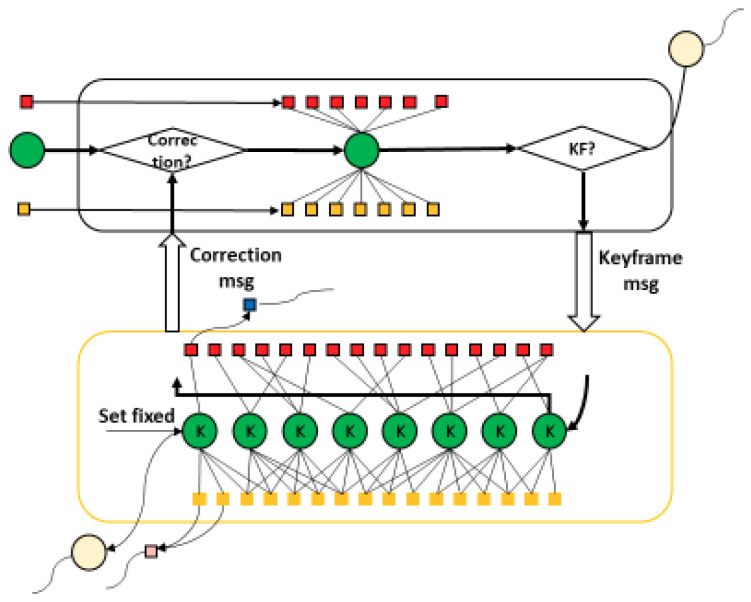
Data communication between tracking thread and local mapping thread.

**Figure 6 sensors-20-04984-f006:**
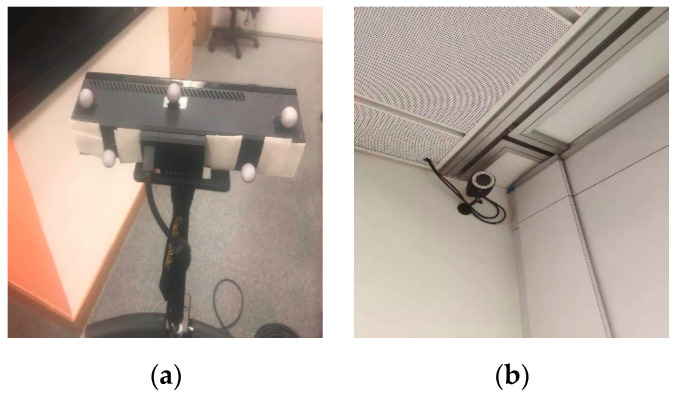
Experiment device. (**a**) Kinect v2; (**b**) Qualisys, the motion capture system.

**Figure 7 sensors-20-04984-f007:**
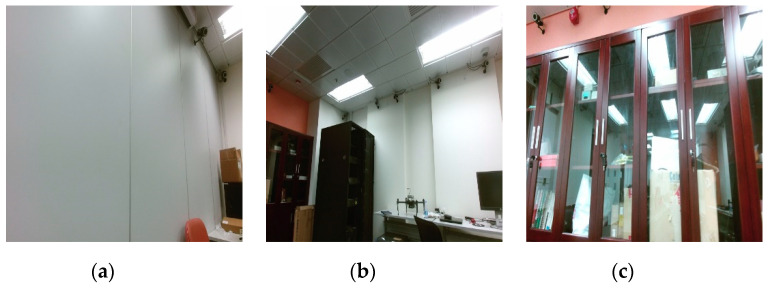
Challenging scenes. (**a**) Low texture; (**b**) Illumination variation; (**c**) Glass window.

**Figure 8 sensors-20-04984-f008:**
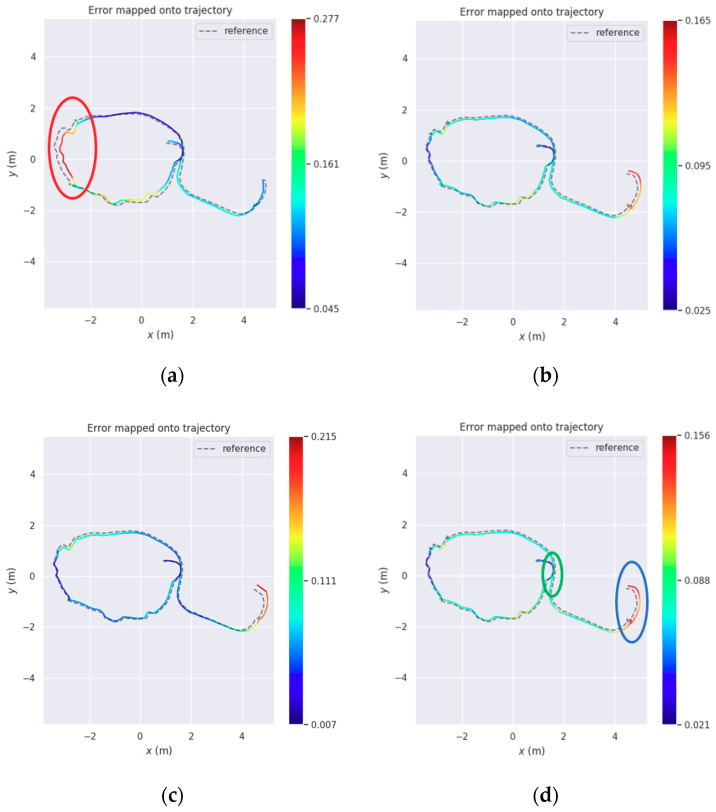
Comparison of estimated trajectories with ground truth on room sequence. (**a**) Point; (**b**) Point + 3D Line; (**c**) Point + 2D Line; (**d**) Proposed.

**Figure 9 sensors-20-04984-f009:**
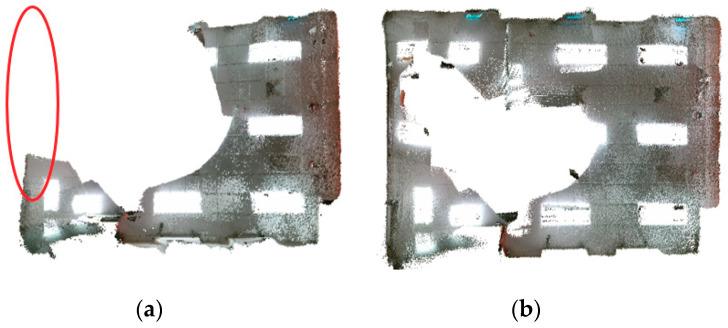
Reconstruction 3D models from simultaneous localization and mapping (SLAM) systems (**a**) ORB-SLAM2; (**b**) Proposed.

**Figure 10 sensors-20-04984-f010:**
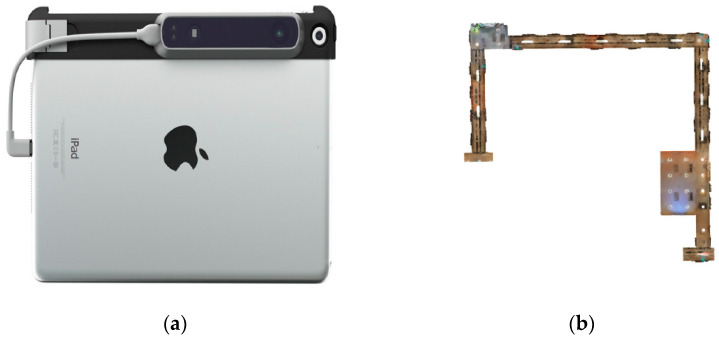
Experiment device and ground truth. (**a**) iPad with a structure sensor; (**b**) Ground truth by NavVis M6.

**Figure 11 sensors-20-04984-f011:**
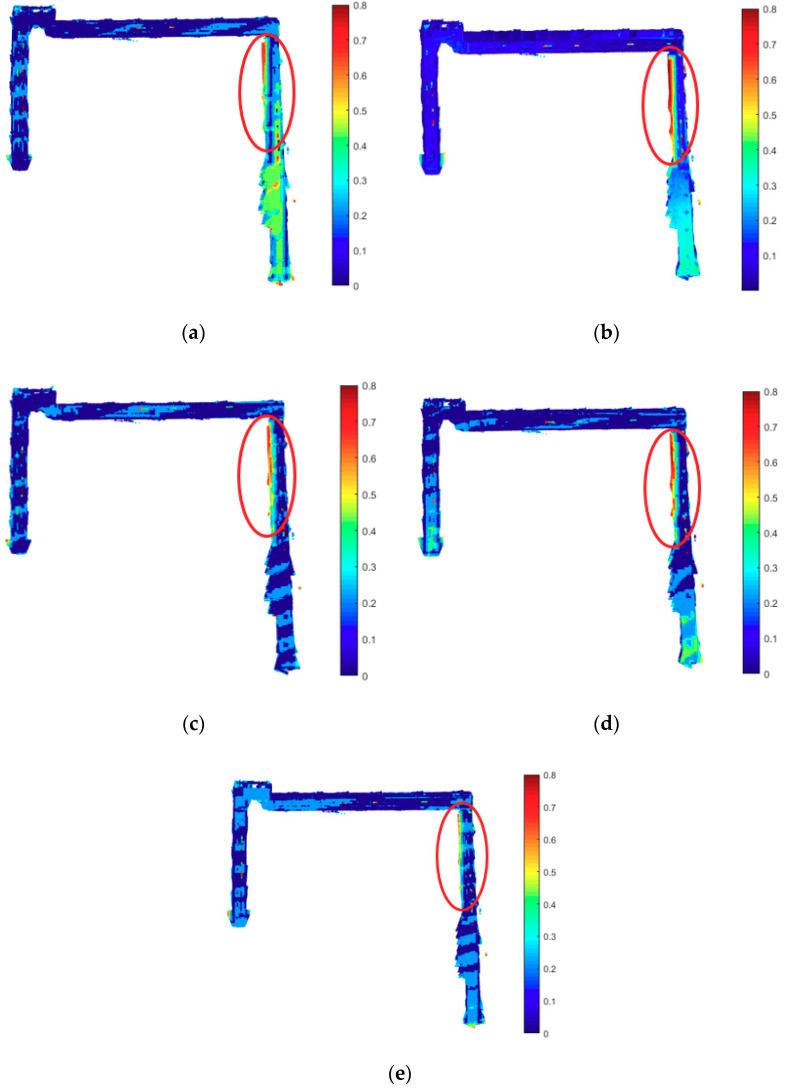
Point-to-point distances between reconstruction models and ground truth. (**a**) ORB-SLAM; (**b**) Point; (**c**) Point + 3D Line; (**d**) Point + 2D Line; (**e**) Proposed.

**Table 1 sensors-20-04984-t001:** Comparison of ATE RMSE (cm) on TUM RGB-D datasets.

Sequence	Length (m)	Proposed	ORB-SLAM2	DVO-SLAM	LSD-SLAM	DSO	PL-SLAM	Canny-VO
fr1_desk	9.3	4.6	**2.1**	2.4	10.7	-	3.0	4.4
fr1_floor	12.6	3.2	6.1	X	38.1	5.5	3.0	**2.1**
fr2_desk	18.9	4.5	1.7	1.7	4.5	-	**1.4**	3.7
fr3_long_office	21.5	6.5	4.1	**3.5**	38.5	14.4	-	8.5
fr3_nstr_tex_far	4.3	7.0	5.6	2.8	18.3	4.8	-	**2.6**
fr3_nstr_tex_near	13.5	**3.3**	3.5	7.3	7.5	3.6	3.5	9.0
fr3_str_ntex_far	4.4	9.0	-	3.9	14.6	18.4	-	**3.1**
fr3_str_ntex_near	3.8	3.7	-	**2.1**	-	-	-	-
fr3_str_tex_far	5.9	1.3	1.3	3.9	8.0	7.9	**0.9**	1.3
fr3_str_tex_near	5.1	**1.2**	1.4	4.1	**-**	24.1	2.6	2.5

**Table 2 sensors-20-04984-t002:** Comparison of ATE RMSE (cm) on room sequence.

Sequence	Length (m)	ORB-SLAM2	Point	Point + 3D Line	Point + 2D Line	Proposed
room	28.4	-	14.9	9.3	9.7	**7.2**

**Table 3 sensors-20-04984-t003:** Comparison of RMSE (cm) of point-to-point distance on corridor sequence.

Sequence	Length (m)	ORB-SLAM2	Point	Point+3D Line	Point + 2D Line	Proposed
corridor	60.8	27.2	30.1	23.8	25.1	**21.4**

**Table 4 sensors-20-04984-t004:** Processing time (ms) of each part of the proposed system and comparison with ORB SLAM2.

Thread	Part	Proposed	ORB-SLAM2
Tracking	Optical Flow	7.5	
	Line Extraction	42.1	
	Robust Pose Solver	2.3	
	Shi-Tomasi Detection	7.6	
	IIR Filter	1.0	
	Total	60.4	32.1
Local Mapping	Local BA	115.3	186.3

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
