# Peer review of "Robust RGB-D SLAM Using Point and Line Features for Low Textured Scene"

_sensors, 2020, doi:10.3390/s20174984_

Round 1

Reviewer 1 Report

This paper proposes a robust RGB-D SLAM system to handle low-texture scenes. The system consists of two main parts: camera pose estimation with extracted point and line features; local Bundle Adjustment for mapping and a bag-of-word approach for loop closing. The key contribution of the system is utilizing both 2D and 3D line re-projection error in the optimization for camera pose eastimation, which brings higher accuracy in low-textured scenes compared with point-based SLAM systems. The paper tries to overcome the problems (e.g. drift of camera poses, tracking failure) in low textured scenes, by considering using both point and line features when optimizing the camera poses in the frontend. The main issues of the paper are that the technical novelty is not sufficient and the experiments are not adequate.  First, integrating line features into the SLAM system is not new, as discussed in the paper, there are several previous works that have utilized line information for SLAM. However, there is no comparison with these line-based SLAM systems. Second, as for handling low textured scenes, which the authors claimed as a contribution, there are also many works that have achieved high-quality results in these kind of sences, e.g. LSD-SLAM[4] and DSO[5]. However, neither a discussion nor an experimental comparison of these works has been conducted. Therefore, the authors are expected to include more discussions and experimental results for line-based SLAM and monocular direct SLAM.   Overall, the exposition is clear, and the writing is not difficult to understand. However, there are still some confusing pieces that need to be clarified: L41: The concept “partial line constraints” is confusing and I do not know what does “partial line constraints” actually mean. It seems that the authors mention it as a novelty of the paper, but I cannot find further explanations or descriptions of this concept in the paper. L212, L232, L241: The definition of “reliable depth measurements” is not clear. Besides, there are some typos:  L28: Augment -> Augmented
L50: Compare -> Compared
L54: Describes -> Describe
L70: Niener -> Nießner
L103: Missing authors before the citation.
L120: Last -> Previous
L134: RGB and color -> RGB and depth
L144: fame -> frame
L220: Pi is not a good notation to represent a parameter.
L327: Gaussian Newton -> Gauss-Newton
L369: variety of -> a variety of
L458: show -> shows
L466: mor -> more
  Missing relevant references: [1] Dai, Angela, et al. "Bundlefusion: Real-time Globally Consistent 3d Reconstruction Using On-the-fly Surface Reintegration." ACM Transactions on Graphics (ToG) 36.4 (2017): 1.
[2] Henry, Peter, et al. "RGB-D mapping: Using Kinect-style depth cameras for dense 3D modeling of indoor environments." The International Journal of Robotics Research 31.5 (2012): 647-663.
[3] Schops, Thomas, Torsten Sattler, and Marc Pollefeys. "BAD SLAM: Bundle Adjusted Direct RGB-D SLAM." Proceedings of the IEEE conference on computer vision and pattern recognition. 2019.
[4] Engel, Jakob, Thomas Schöps, and Daniel Cremers. "LSD-SLAM: Large-scale direct monocular SLAM." European conference on computer vision. Springer, Cham, 2014.
[5] Engel, Jakob, Vladlen Koltun, and Daniel Cremers. "Direct sparse odometry." IEEE transactions on pattern analysis and machine intelligence 40.3 (2017): 611-625.

Author Response

Dear Reviewer,

Thanks a lot for giving us an opportunity to revise our manuscript.

Our response letter is attached.

Best regards,

Yajing Zou

Reviewer 2 Report

This paper focuses on feature-based SLAM and particulary on point and line features in low textures scenes.

Introduction and Related works parts are well done and the citations are appropriate.

The part "3. System Overview and Notation" is quite clear and classical ... perhaps a little bit short.

The part "4. Frontend" is a key part of the methodology and the part "4.1 Feature extraction and matching" is a lockdown for the proposed method. This part is really too short. The authors simply use very classical features extractors for points and lines. Those algorithms are time consuming (in particular LSD / LBD) and not really efficient in low texture images. This part really needs to be more convincing on this kind of images!

Moreover the authors give some experimental values without any justification. For example:
- Why the image plane is divided in 16 regions? Doesn't it depends on image resolution?
- How the wrong matches are managed? Could you explain cross-check and ratio-test? Which value is use for ratio-test?
- Why 0.8 is a good parameter for IIR filter? How is it correlated with the quality of the features descriptors?
- What is the threshold for RANSAC step?
- Which "constant-velocity motion model" is used?
- ...

The part "5. Backend" is well written. The authors give again some experimental values. It should be smart to justify them.

The authors said that "This paper focuses on feature-based SLAM in consideration of real-time performance." (line 90) ... but it seems that the approach needs a very low frame rate to be "real time".

Author Response

Dear Reviewer,

Thanks a lot for giving us an opportunity to revise our manuscript. 

Our response is attached.

Best regards,

Yajing Zou

Round 2

Reviewer 1 Report

It is good to see that all my concerns have been addressed in the revised version. 

Reviewer 2 Report

The authors have addressed most of the comments. Thanks. There is still some "experimental values" that are not well justified and should be improved to achieve the study ... but the paper is well written and the methodology is clear. So I thing the paper can be published.